# Integrated Management Strategies for Viral Nervous Necrosis (VNN) Disease Control in Marine Fish Farming in the Mediterranean

**DOI:** 10.3390/pathogens11030330

**Published:** 2022-03-08

**Authors:** Francesc Padrós, Massimo Caggiano, Anna Toffan, Maria Constenla, Carlos Zarza, Sara Ciulli

**Affiliations:** 1Departament de Biologia Animal, de Biologia Vegetal i d’Ecologia and Servei de Diagnòstic Patològic en Peixos, Universitat Autònoma de Barcelona, 08193 Barcelona, Spain; maria.constenla@uab.cat; 2Panittica Italia S.r.l., 72015 Fasano (BR), Italy; m.caggiano@panitticaitalia.it; 3National Reference Laboratory for Fish Diseases, OIE Reference Laboratory for Viral Encephalopathy and Retinopathy, Istituto Zooprofilattico Sperimentale delle Venezie, 35020 Legnaro (PD), Italy; atoffan@izsvenezie.it; 4Health Department, Skretting Aquaculture Research Centre, P.O. Box 48, 4001 Stavanger, Norway; carlos.zarza@skretting.com; 5Department of Veterinary Medical Sciences DIMEVET, Alma Mater Studiorum University of Bologna, 47042 Cesenatico (FC), Italy; sara.ciulli@unibo.it

**Keywords:** viral disease, sea bass, sea bream, viral encephalopathy and retinopathy, biosecurity, disease management

## Abstract

Viral nervous necrosis (VNN) is the most important viral disease affecting farmed fish in the Mediterranean. VNN can affect multiple fish species in all production phases (broodstock, hatchery, nursery and ongrowing) and sizes, but it is especially severe in larvae and juvenile stages, where can it cause up to 100% mortalities. European sea bass has been and is still the most affected species, and VNN in gilthead sea bream has become an emerging problem in recent years affecting larvae and juveniles and associated to the presence of new nervous necrosis virus (NNV) reassortants. The relevance of this disease as one of the main biological hazards for Mediterranean finfish farming has been particularly addressed in two recent H2020 projects: PerformFISH and MedAID. The presence of the virus in the environment and in the farming systems poses a serious menace for the development of the Mediterranean finfish aquaculture. Several risks associated to the VNN development in farms have been identified in the different phases of the farming system. The main risks concerning VNN affecting gilthead seabream and European seabass have been identified as restocking from wild fish in broodstock facilities, the origin of eggs and juveniles, quality water supply and live food in hatcheries and nurseries, and infected juveniles and location of farms in endemic areas for on-growing sites. Due to the potential severe impact, a holistic integrated management approach is the best strategy to control VNN in marine fish farms. This approach should include continuous surveillance and early and accurate diagnosis, essential for an early intervention when an outbreak occurs, the implementation of biosecurity and disinfection procedures in the production sites and systematic vaccination with effective vaccines. Outbreak management practices, clinical aspects, diagnostic techniques, and disinfections methods are reviewed in detail in this paper. Additionally, new strategies are becoming more relevant, such as the use of genetic resistant lines and boosting the fish immune system though nutrition.

## 1. Introduction

### Farmed Species, Production System and Diseases in the Mediterranean

Aquaculture is a continuously growing sector, with an average annual growth of 4.5 percent per year in the 2011–2018 period and reaching the all-time record high of 114.5 million tonnes (t) production in live weight in 2018 with a total farmgate sale value of USD 263.6 billion [1]. The total production of aquatic animals consisted of 82.1 million t, with a value of USD 250.1 billion, representing the 46 percent of the global food fish production in 2016–2018 [1]. Perspectives make provision for further increase in fish production; food fish consumption is also expected to increase in all regions and subregions by 2030 in comparison with 2018. The role of aquaculture in satisfying this demand is essential; indeed, the major growth in production is expected to originate from fish aquaculture, which is projected to reach 109 million t in 2030, with growth of 32 percent over 2018 [1]. Finfish marine aquaculture production in the Mediterranean is mainly dominated by gilthead sea bream (*Sparus aurata*), European sea bass (*Dicentrarchus labrax*) and meagre (*Argyrosomus regius*) [2] although some other species (pagrus, amberjack, mugilidae, etc.) are produced in a lesser extent. European production had reached 208,197 t for European seabass and 199,476 t for gilthead seabream in 2019 [3] and other Mediterranean countries, such as Egypt (24.498 t sea bass, 36,000 t sea bream) and Tunisia (3.500 t sea bass, 16.000 t sea bream) undertake significant production of both species [4].

The farming production scheme is similar in all these species and is based on obtaining eggs from specific broodstocks, egg hatching and larval rearing in hatcheries, post-larval weaning and juvenile rearing in nurseries, pre-ongrowing until juveniles reach a certain weight and are ready to be transferred to the final ongrowing farms, usually in sea cages but also in some cases in inland facilities. Due to the length and technological complexity of the early phases of their life cycle (complex sexual maturity induction, continual spawning, necessity of different live prey for the early stages, weaning, etc.), many Mediterranean aquaculture facilities are highly specialized as hatcheries.

The structure of the Mediterranean fish farming industry is characterised by several specific elements. The first one is the complex scenario, where 22 countries from three different continents (Europe, Asia and Africa) develop aquaculture activities in a single common area. Some of these countries are also members of political and economic unions such as the EU. In addition, some of these countries develop fish farming activities in other areas (Atlantic Ocean, Red Sea, Black Sea). Another element is the structure and size of the farming companies. Some of them are international companies (operating in two or more countries) and some other are based only in a single country. Some of them are large aquaculture companies, with more than 2000 employees, with different facilities (hatcheries, nurseries, ongrowing sites) in many different places, some of them are medium size, with several production units and facilities and other farms are smaller, with one or two production units. In some cases, companies operate following closed farming schemes (the company owns facilities with all the different production phases) and in other cases, farming companies are highly specialised in only one part of the production cycle (hatchery, nursery, pre-ongrowing or ongrowing).

Amongst the main problems and limitations, diseases are still the main bottlenecks for the full development of fish farming in the Mediterranean [5,6,7,8]. Amongst diseases, viral nervous necrosis (VNN), also known as viral encephalopathy and retinopathy (VER), is one of the most relevant ones, with a huge impact in the industry due to the serious losses and the disease-associated burdens. VNN is caused by nervous necrosis viruses (NNVs), small single stranded positive sense RNA viruses included in the genus *Betanodavirus*, *family Nodaviridae*. Betanodavirus virions are non-enveloped, roughly spherical in shape, 25–33 nm in diameter and have icosahedral symmetry. The Betanodavirus genome consists of two molecules of RNA: the RNA1 which encodes the RNA-dependent RNA polymerase (RdRp) and RNA2 which encodes the coat protein (CP); moreover, a subgenomic transcript called RNA3, originating from the 3’ terminus of RNA1, is generated during virus replication [9].

Genetic analysis clusters betanodaviruses in four genotypes currently accepted by the International Committee on Taxonomy of Viruses (ICTV) as official species of this genus: Striped jack nervous necrosis virus (SJNNV), Tiger puffer nervous necrosis virus, Barfin flounder nervous necrosis virus and Redspotted grouper nervous necrosis virus (RGNNV; [9]). Moreover, reassortant strains have emerged from the reassortment between RGNNV and SJNNV genotypes: the RGNNV/SJNNV inheriting the RNA1 from the RGNNV genotype and the RNA2 from the SJNNV genotype and the SJNNV/RGNNV inheriting the RNA1 from the SJNNV genotype and the RNA2 from the RGNNV genotype [10].

## 2. VNN and Species

VNN outbreaks and betanodavirus presence has been detected in a large number of fish species all around the world [11]. Particularly in the Mediterranean, European sea bass is the species with the highest number of references of VNN reports associated with betanodavirus identifications, despite different conditions (i.e., salinity, age, temperature, viral genotypes) it is considered the most susceptible and sensitive species in the Mediterranean Sea [12]. However, other farmed species in the Mediterranean such as seabream, shi drum (*Umbrina cirrosa*) and mugilidae have also been involved in clinical VNN outbreaks. Outbreaks in other species present and farmed in the Mediterranean such as amberjack (*Seriola dumerili*) and several grouper species have also been reported in other geographical areas. Senegalese sole (*Solea senegalensis*) is also a very susceptible species to VNN [13] and many studies on VNN in this species have been published [11]. Senegalese sole is also present and occasionally farmed in the Mediterranean but is mainly commercially produced in the Atlantic area. Other flatfish including common sole (*Solea solea*) and turbot (*Scophthalmus maximus*) have been reported to be highly susceptible [14,15].

European sea bass can be clinically affected during their entire life cycle, with the highest mortality rates in larvae and postlarvae, but also at a relevant extent in juveniles and during all the production cycle. Seabream was traditionally considered as a non-susceptible species and therefore, regarded as an alternative species in certain areas with high pressure of VNN in seabass. However, some case reports [16] described sporadic episodes of mortality apparently associated to VNN in seabream juveniles reared in cages and occasional detection of betanodavirus in seabream in routine health surveys by molecular techniques was not uncommon. More recently, severe outbreaks of VNN in larvae and young juveniles were recorded in different places and caused by the reassortant betanodavirus RGNNV/SJNNV [17,18,19].

In any case and so far, gilthead seabream susceptibility to VNN seems to be restricted to larval stages and not to juveniles or bigger sizes and is associated with the presence of the reassortant strains [12,19]. Clinical outbreaks in meagre are not recorded for the time being [20], although RGNNV and SJNNV betanodavirus co-infections have been detected in wild populations [21]. As outbreaks associated with VNN have been described in several species of Sciaenids (i.e., shi drum) and due to the progressive increase in the production of meagre, it would not be surprising if VNN could also affect meagre production in the future.

The increasing emergence of betanodavirus reassortants in recent years substantially modifies the past scenario in the Mediterranean area, where each NNV genotype was associated with specific fish species and thermal profiles, introducing a greater uncertainty about how VNN would evolve in the future in the different species, ages and environments.

## 3. VNN and Betanodavirus in the Wild

Betanodavirus are widely present in wild fish populations in the Mediterranean Sea (see list of species reported in [11]). Relevant mass mortalities associated to VNN have been described particularly in dusky grouper (*Epinephelus marginatus*) [22,23,24,25] and most recently this virus is also suspected to be associated with moray eel (*Muraena helena*) mortality [26].

Betanodavirus were also detected in marine invertebrates using molecular techniques [27,28,29,30]. The potential risk associated with the presence of betanodaviruses in marine invertebrates such as bivalves or polychaetes should not be disregarded but, in this case, invertebrates should be considered as vectors, potentially associated with filtration activity of some of these organisms [31]. The detection of betanodavirus in other aquatic organisms such as cephalopods and their role as vectors and reservoir hosts has also been described [32]. The presence of betanodavirus in these organisms is a risk to be considered particularly at hatchery level, as sometimes squid or mussels are used to feed broodstock fish.

## 4. VNN in the Farming System

VNN can affect all the different phases in the production scheme but the risks, the expression of the disease, the impact and the consequences can be different in each phase and depend on many other factors. Maybe one of the most evident and shocking effects of acute VNN outbreaks are the catastrophic consequences seen in hatcheries with massive mortalities up to 100% of the stock or in cages with cumulative mortalities up to 35% in 35 g seabass and 25% in 150 g sea bass (Zarza, personal observations). In addition, the consequences of the presence of the virus in the facilities go beyond the impact of the mortalities. Even if there are no mortalities or the fish does not display any signs of VNN, the detection of betanodavirus-positive fish in a seabream or seabass broodstock facility or the detection of positivity in a single batch of larvae, postlarvae or juveniles may put in serious jeopardy the whole hatchery or nursery. In the case of virus-positive juvenile stocks, viral detection even in the absence of mortality should be grounds for preventing movement and obviously they should not be sold. Consequently, this virus-positive stock becomes a significant logistic and health problem for the farm and should be immediately sacrificed due to risk of spread of NNV. The presence of a betanodavirus-positive batch of larvae can have also devastating effects, as this usually forces the hatchery to establish radical measures, including the destruction of the whole batch of larvae, the application of strict disinfection measures, intense monitoring of the subsequent batches and consequently limiting the production capacity of the facility for several weeks. The situation can be even worse if the broodstock also tests positive for betanodavirus, as this can compromises the viability of the future production batches and the investment in genetic selection performed.

In seabass and to a lesser extend in seabream reared in cages or ponds, the impact in terms of morbidity and mortality is highly variable and depends on many different factors such as virus strain, water temperature, mixed infections, fish husbandry and vaccination, amongst others. In cages, VNN outbreak duration is usually 4 to 6 weeks, with high mortality and severe reduction in feed intake and consequently growth. Affected batches can have poor growth for weeks and even for several months after the onset of the infection. This is an effect that is not always considered in the calculation of the disease impact. The situation could become even worse, as frequent mixed bacterial infections with *Vibrio anguillarum*, *Vibrio harveyi* or *Photobacterium damselae* subs. *piscicida* are usually diagnosed at the same time [33], as these diseases are usually expressed during the same risk period (summer-autumn) and can appear as a consequence of the debilitation of the fish. In this case, there is a higher impact and a worse prognosis due to the frequent loss of efficacy of oral treatment with antibiotics and a longer duration of outbreak.

VNN must also be considered as a hazard when a new species is studied for exploitation as promising opportunity for the market. Relevant examples of the negative impact of NNV in the past were shi drum and red drum (Toffan personal communication) and cod and halibut [34,35].

## 5. Epidemiological Aspects of VNN

The widespread nature of NNV in terms of geographical distribution and host range makes this virus a global threat to fish stocks. Indeed, NNV is present in all parts of the world where aquaculture is practised. Currently, VNN has been reported in almost all Mediterranean countries involved in European sea bass and gilthead sea bream production: Spain, France, Italy and Greece, as well as in North African countries [11] and has been considered for many years the most important viral problem for Mediterranean finfish farming. VNN outbreaks were detected for the first time in many Mediterranean European seabass on-growing farms in the summer 1995, with increasing mortality in the 1995–1998 period. Since then, VNN has been considered a real threat to Mediterranean aquaculture [36] and currently it is considered the major limiting factor in sea bass production through Mediterranean countries and a new challenge in sea bream farming. Recent consultations among scientific fish health experts [37] and fish producers’ associations (H2020 European Project PerformFISH deliverable 3.2, [8]) ranked VNN as the disease with the greatest impact in the Mediterranean European seabass production.

The widespread presence of betanodavirus in the marine aquatic environment represented by the detection of the virus in many wild fish species, including escapees and also in marine invertebrates is a key epidemiological aspect to understand the difficulties to manage the disease. For the current farming scenario in the Mediterranean based mainly on cage production, farmers should assume that farmed fish are reared in an environment with a potential exposure to the virus and, therefore, are at a real risk of becoming infected by betanodavirus and developing VNN. The close relationship of the viral sequences between the isolates collected in wild affected fish and those isolated during clinical disease outbreaks in farmed fish in the same area in previous years suggests persistent circulation of betanodaviruses and transmission between wild and farmed stocks [22]. However, VNN has been also found in area without direct water connection with finfish farms rearing susceptible species suggesting that VNN infection could complete its epidemiological cycle in the natural environment [38]. The great potential diffusion of virus into wild fish [39] makes even the potential impact of this infection on wild fish stocks even greater. Furthermore, as the disease in the Mediterranean area is mainly associated with warm temperature, the climate change could increase the impact of the NNV switching asymptomatic infections into clinical outbreaks [40].

The level of exposure to the virus and the risk of developing a VNN outbreak may change according to different factors, such as the prevailing genotypes in each area, outbreak recurrence in a certain area, temperature profiles, water current or concentration of farms and cages in the same area, amongst others. In this scenario, the risk to face problems associated with VNN can be managed in different ways. For broodstocks, hatcheries and nurseries, biosecurity is the best strategy, meanwhile in ongrowing stages, vaccination and management are the keys to reduce the potential impact of the disease. All these aspects will be commented in more detail in the following sections.

## 6. Economic Impacts of VNN

Despite precise information or published information on global economic impact associated with VNN in the Mediterranean aquaculture not being available, some considerations and data can be provided to convey the impact of this disease. Nevertheless, the disease burden and the economic impacts of VNN could be different in the different production phases.

The impact of VNN has been particularly relevant to hatchery and nursery production and especially in European sea bass and following the emergence of the RGNNV/SJNNV reassortant strain, VNN has also affected hatchery production of gilthead sea bream [17]. Outbreaks of VNN in hatcheries typically result in termination of the affected production batch on the assumption that mortality will be very high with values around 100% in the larval stage and 85–90% in the weaning phase. When mortality does not reach 100%, euthanizing survivors should be seriously considered. Farming of survivor fish is not economically convenient nor epidemiologically advisable as the risk of having asymptomatic carrier fish in the survivor stock is very high. Hatcheries are requested to provide batches of healthy and pathogen-free juveniles for ongrowing, so the suspicion of the circulation of the virus in hatcheries even in the form of subclinical infections not only limits the value of juvenile fish but also is unacceptable from the quality assurance point of view and may induce a negative image for the facility or public image for the company.

The development of VNN outbreaks in market-size fish batches is also relevant in Mediterranean aquaculture for European sea bass as this species has been shown to be sensitive to NNV in a wide range of ages (see section in farming system). The impact on the ongrowing phase is often severe, even if, at this stage, mortality is lower. In European sea bass, land-based and in cages, cases with 30–50% of cumulative mortality in a single batch have been observed after recurrent outbreaks, but generally from 5% in bigger fish to 30–40% mortalities in juveniles, rearing conditions and virus strain are reported by farmers and the literature [41]. Farmers reported that an accumulated mortality rate over 25% makes one sea bass batch production not cost effective. A recently published paper based on the MedAID project proposes an approach to evaluate the impact as direct costs of diseases caused by different pathogens, with a particular focus on VNN, in Mediterranean sea bass grow-out farms [7]. In addition to the economic value of the mortalities in the context of the production batch, it is also very important to consider the extra costs of resources, materials and personnel associated with the collection, management, and disposal of the mortalities, as it is described in more detail in the outbreak management section.

As it was previously indicated, seabass survivors of VNN outbreaks in cages or ponds present a higher dispersion of size and weight in the stock and display reduced growth and zootechnical performance during the following months. This is an added economical handicap for the farmers as the batches previously impacted by VNN outbreaks will not recover their growth potential and most probably they will be a source of economic losses. For this reason, it is highly advisable to consider earlier harvesting of these cages or ponds if fish have reached market size and considering the potential economic value in the market of these fish. If not, culling the stock and start a new production batch can be an alternative to reduce economic losses.

Conversely, the situation in gilthead sea bream is very different as, until now, no significant outbreaks in ongrowing gilthead sea bream have been recorded. For many years, sea bream was considered a non-susceptible species for VNN and this apparent resistance was used sometimes as a management tool to reduce the impact of VNN outbreaks. In the past, simultaneously rearing cages with sea bass with cages of sea bream was seen as a strategy to reduce the total biomass of the susceptible sea bass. Sometimes, when VNN outbreaks hit farms with sea bass the previous year, farmers decide the following year to produce sea bream instead of seabass as an alternative to reduce the virus circulation. The situation is nowadays very similar and no relevant VNN outbreaks are currently detected in sea bream. However, some cases with mild mortalities in sea bream batches with NNV-positive fish as well as NNV-positive fish in larger sea bream batches with no mortality are occasionally recorded during routine health controls. Recently, interspecies transmission between seabream and seabass has also been reported for the RGNNV/SJNNV reassortant strain [18]. In this scenario it is difficult to predict if the situation will be the same or could change in the future.

In addition to all these impacts, it is important to bear in mind that insurance can mitigate the economic risks associated to the occurrence of diseases in fish farms. Aquaculture insurance typically covers losses associated to different hazards including environmental issues such as storms, contamination, accidents, theft, effects of predators, malicious damage and diseases. Each insurance company has different policies about how diseases and economic losses related to diseases are covered and, particularly, some companies consider VNN outbreaks as a relevant hazard for marine farming in the Mediterranean and, consequently, a risk that should be adequately addressed in the specific insurance conditions or in further extensions as well as evaluated in the calculation of the insurance fees or in their own policies. It is clear that if farms that do not take into consideration the specific risks of VNN in their facilities and do not invest in biosecurity and in preventive (mainly vaccination when possible) and mitigation measures, they are at a higher risk of suffering heavy economic losses. Thus, this document is also intended to provide to the insurance experts working on aquatic farm insurance some recommendation on how to assess the real risks of VNN for each farm.

Concerning VNN outbreaks, it is also important to remark that VNN is not considered or listed in the current EU legislation as an important disease. Furthermore, VNN is not under any specific surveillance programme so, in other words, detection of VNN does not result in official economic compensation mechanisms by the authorities, and this is one of the main reasons why is highly recommended to include VNN in the insurance coverage.

## 7. VNN Risk Assessment

As described above, VNN can affect all the different phases in the production system, but the potential risks and their relevance can be different in each of them. For this reason, an exercise to identify the main risks concerning VNN affecting gilthead seabream and European seabass in each of the production phases (broodstock, hatchery, nursery and ongrowing) was developed within the framework of the PerformFISH European project. This first exercise involving scientists specialized in fish health, and specifically in the Mediterranean aquaculture, was characterised by a HAZOP (Hazard and Operability Study) evaluation system, combined with the information available from technical and scientific literature, as well from their personal experience (see Deliverable 3.4, PerformFISH, [42]). From this first internal assessment, a more complete and detailed evaluation on the relevance of each risk factor, including assessment by experts on the field, was develop by using DELPHI techniques.

Based on these results, having dedicated well-trained personnel and with high awareness of well-established biosafety protocols was considered the most relevant factor to avoid the appearance of VNN in a facility or the presence of NNV or outbreaks. This recommendation was common to all the production phases and virtually all the consulted experts agreed on that. Concerning risks, the relevance of the different risks evaluated were different in the different production phases. 

### 7.1. Specific Risks in Broodstock Facilities

In the past, many hatcheries used fish selected from commercial batches with good production performances as a main system of generational replacement and genetic improvement. Additionally, some hatcheries used fish collected from the wild to retain “wild” morphological traits in the offspring. All the experts agree that the supplementation of broodstock with wild fish was the riskiest practice for VNN. These procedures were progressively phased out due to the high sanitary risks involved and the implementation of new genetic selection techniques (microsatellites, SNPs). Currently, most of the seabream and seabass broodstocks consist (or should consist) of fish from the same or from different origins but always reared in controlled conditions and supervised with similar sanitary protocols.

The quality of the water supply is another important risk factor addressed in broodstock. Water pumped from open sea or, especially, from shallow areas or lagoons are the worst option for keeping a facility safe from VNN. In addition to the strategies related to control the origin of the fish and water, there was a high consensus on recommendations to rear all broodstock populations under a health management plan, which should include an accurate surveillance program for detection of asymptomatic carriers. According to experts, the most reliable and effective diagnostic techniques for the detection of VNN are molecular tests (RT-PCR and RT-qPCR) on broodstock sperm and eggs, as well as in different tissues of sentinel fish, followed by further testing of the batches of larvae and juveniles originated from this broodstock. Regarding management strategies within the farm to avoid problems with VNN, most experts highlighted the need for high level of sanitization, which includes immediate disinfection of the tools (nets, egg collectors) after each use and especially trained staff in biosafety protocols.

### 7.2. Specific Risks in Hatcheries

Although gilthead seabream and European sea bass larvae, like other similar larvae, are delicate organisms and very susceptible to physical, chemical and biological challenges and hazards, it is striking that almost none of the options highlighted in the evaluation (eggs origin, larvae batch testing, food, facility design, operational procedures) were really underlined by the experts in Delphi as paramount risks. Following a range from the most likely risks which could cause future problems associated to VNN, the origin of the eggs, the water quality and live food were identified as the most relevant. Eggs from facilities where broodstocks are not tested for NNV or eggs from an external supplier with no specific VNN-health certification were identified as the main risk factors. The absence of water treatment (filtration combined with UV or ozone) was also identified as a relevant risk and the use of non-certified NNV-free microalgae during the larval rearing and the acquisition of artemia without NNV-free certification or inadequate cyst hatching protocols were recorded as the worst options concerning VNN risk. The most important factor to avoid NNV infections in a farm, in addition to training staff in biosafety protocols, was to routinely test larvae batches for NNV and to rear them separately, avoiding mixing batches according to hatchery requirements.

### 7.3. Specific Risks in Nurseries

During this period, the highest risks for VNN infection were related to the water supply, the origin of the post-larvae and the reared system. Water pumped from the sea or from shallow areas or lagoons are again the worst options in terms of risk, as well as the placement of untreated water effluents close to the general water intake and without applying any treatment. To avoid NNV infections, the better options highlighted by the experts are to acquire certified NNV-free juveniles from the same source, with transport systems also managed by the same company, and routinely testing these fish during nursery and pre-ongrowing. There was also consensus that, in nurseries, recirculation systems provide a higher biosecurity level. The same recommendation can be also applied for broodstock and hatchery units, where the biosecurity management in RAS is more manageable due to the lower water renewal required.

### 7.4. Specific Risks in Ongrowing Facilities

Different systems are used for the ongrowing phase in European sea bass and gilthead sea bream production: sea cages in sheltered areas or offshore conditions and land-based facilities based in tanks or ponds. Each system has different characteristics, requirements and sanitary risks that were assessed, but in both (cages and land-based facilities), origin and health quality of the juveniles was the main scored risk. Most experts agreed that the risk of infections was higher if juveniles brought into the facility were not certified NNV-free, especially if they came from different companies. In addition, the risk in ongrowing in tanks and ponds had risk scores similar to nurseries. The highest score on VNN sanitary risks was related to the location of the cages. Farms located in a VNN endemic area, with recurrent outbreaks detected in the same farm or other farms around 25 miles in previous years, and with water temperatures above 25 °C (especially if these temperatures are sustained over time for more than 2 or 3 months) were considered as the scenario with highest VNN risk. On the other hand, one of the most beneficial practices to avoid VNN infections in both production systems was the systematic vaccination with VNN commercial vaccines following the manufacturer’s recommendations.

## 8. Betanodavirus Surveillance

Concerning finfish diseases and disease surveillance, it is important to consider that in the EU, a new regulation entered into force on 21 April 2021. This legislation is based on the EU’s Animal Health Law (Regulation (EU) 2016/429 on transmissible animal diseases, [43]) and includes the regulations supplementing or implementing the EU’s Animal Health Law, that includes, amongst others, delegated Regulation (EU) 2018/1629 on listing of diseases.

One of the pillars of this new legislation is the prevention of emerging diseases. Therefore, all these regulations should be considered as a legal basis for the authorities to monitor the impact on any mortality events whatever the fish species affected and as a tool to avoid further spreading of infectious diseases.

It is important also to highlight that, in other non-EU European countries, other regulations apply, but the EU’s Animal Health Law can be considered as an example of wellupdated regulation based on risk assessment evaluation and the OIE recommendations.

Concerning betanodaviruses and VNN, it must be stressed that VNN has been recently removed from the OIE Diagnostic Manual for aquatic animals. VNN is a disease that is not under any current general surveillance programmes at the EU country level. This is mainly due to the fact that VNN is considered ubiquitous in marine waters, is not zoonotic and no country has never claimed to be free from the disease. However, this does not mean that VNN is not a relevant disease. In fact, the seabream and seabass farming industry in the Mediterranean is very aware and sensitive of the risk posed by this disease, and the EU is also attentive to potential risks as a result of the evolution of VNN. This is the reason why, amongst others, betanodavirus and VNN are some of the main targets of the research on fish diseases in Europe and VNN is usually considered in H2020 projects and in national research projects, and also farms and farming companies have developed and implemented their own surveillance programs. Hatcheries and nurseries tend to develop and apply strict surveillance programs included in more extensive health control plans and structured following similar quality control criteria. Most of the batches of juveniles are routinely tested one or more times during rearing, and often just before juveniles are delivered to the ongrowing site. However, not all these surveillance programs are homogeneous in design and have the same levels of confidence and there are still differences in methods or strategies between hatcheries, but the general trend is towards a progressive implementation and improvement of these programmes. Concerning VNN-specific sampling in hatcheries, a recommended strategy could be to test systematically all larval batches a few days after hatching (e.g., at 7 dph) for early detection and then repeat sampling at least two weeks later and then again one more sampling in postlarval stage and a fourth test at the juvenile stage (around 0.5–1 g). In this way, each batch of juveniles could be tested at least four times during its rearing in the hatchery and nursery period. To target weak or abnormal larvae in the sampling procedure may also increase the virus detection efficiency.

Many laboratories from research centres, universities, health institutions and even health departments of the fish farms have already strong diagnostic and detection capabilities and, recently, the MedAID and PerformFISHi projects made a joint effort to identify the laboratories working in the Mediterranean including their specific diagnostic capabilities on VNN and other diseases. This initiative was also supported by the implementation of two rounds of proficiency testing (years 2018–2020) developed and managed by Istituto Zooprofilattico Sperimentale delle Venezie (IZSVe). In this scenario, it is important to highlight the role played by IZSVe as the OIE VNN reference laboratory. Remarkably relevant for many years was the work of Dr. Giuseppe Bovo leading this laboratory, advising, guiding and supporting fish health specialists in Europe and also passing his legacy to the current IZSVe staff. Now they still have a relevant role in terms of development, testing and validation of diagnostic techniques, provision of standard reagents, training activities as well as continuous support to researchers and other laboratories working in this field.

## 9. Disease Clinical Patterns, Identification and Diagnostic Tools

Typically, VNN outbreaks in European sea bass associated with the RGNNV genotype occurs at high temperature (about 25°C) that in Mediterranean means late summer or early autumn [44]. However, other genotypes have been associated with VNN outbreaks in Southern Europe, including sporadically the SJNNV genotype and more frequently the reassortant strains SJNNV/RGNNV and RGNNV/SJNNV and involving other species such as gilthead sea bream, Senegalese sole, in addition toEuropean sea bass [10,13,18]. These outbreaks occur in environmental conditions, different from those generally described for RGNNV infection in sea bass such as in the case of mortality episodes observed at about 19–21 °C in gilthead sea bream and European sea bass larvae associated to a RGNNV/SJNNV reassortant strain [17,18]. Certainly, the circulation of several betanodavirus strains and the involvement of different fish species makes it necessary to investigate the presence of betanodaviruses with laboratory tests in disease/mortality outbreaks observed in several environmental conditions.

Moreover, the increasing occurrence of *V. harveyi* infection [45,46] associated to the presence of nervous signs in sea bass and bream outbreaks imposed the need for differential diagnosis and for laboratory analysis to confirm aetiology.

VNN is generally expressed with outbreaks of acute mortality; clinical signs consist mainly of behavioural abnormalities and external and internal lesions. Affected fish display altered swimming behaviour—spinning, circular, rapid “bursts” or sprint, upside-down position—and external and internal lesions (brain congestion, swim bladder hyperinflation, exophthalmia and eye damage, from corneal damage to panophtalmitis, due to erratic swimming and injuries due to mechanical damage when fish bump into the nets) that are associated to the disease and makes sick fish easier to detect. However, these signs and lesions are not always present or obvious. In fact, some of the fish involved in VNN outbreaks have no evident gross lesions or do not display evident clinical signs. In addition, many other pathologies and particularly in later stages of septicaemic bacterial problems may have one or several signs or lesions similar to those described in VNN-affected fish. For this reason, these signs and lesions should be considered only presumptive indicators of VNN, and it is highly recommended to apply other diagnostic procedures to increase the diagnostic strength.

In histopathology studies, microscopic findings such as vacuolation and necrosis of nervous cells of the spinal cord, brain and/or retina and gliosis are typically considered as highly suggestive of VNN [17,47]. These lesions can be observed in diseased fish, particularly in larvae and mainly in acute cases, but in many cases these lesions are not so clear or evident and can be misidentified by normal structures, other diseases such as infection by rickettsia or even by post-mortem artefacts, mainly if the pathologist does not have experience in this disease. In this case, it is highly recommended to support these observations using immunohistochemical (IHC) or in situ hybridization (ISH) techniques. If samples are taken from the peak of mortality or in subclinical cases, the value of histopathology is lower, as the lesions are less evident, and the viral load is lower. Confirmatory diagnosis and target surveillance based on cell culture and molecular techniques are advisable in these cases.

The confirmatory diagnosis has to be based on tests that can confirm the presence of the virus in a biological sample collected from suspected fish; on the other hand, the target surveillance must be based on techniques able to detect the virus in a fish population, focusing mainly on the presence of asymptomatic carriers.

Many methods have been developed and used to detect and study NNV infection; however, the practical application of most of them is restricted due to the existence of several limitations.

The use of permissive cell culture followed by immunological or molecular identification represents the reference (Gold Standard) method due to its high sensitivity and the ability to provide a viral isolate for further characterization [48]. Sensitivity assays showed a limit of detection in the range of 10^1.55^–10^1.8^ TCID_50_/mL for virus titration on cell cultures [28,49]. However, this method is time-consuming and requires a great experience and a specialized laboratory to manage cell cultures [44]. Virus isolation is mainly based on the use of two cell lines (SSN-1; E-11) available through the European Collection of Cell Cultures (ECACC) and permissive for the four genotypes [48]; however, other cell lines have been developed and used for research and diagnostic purposes.

Antigen-detection methods, such as the indirect fluorescent antibody test, usually used to identify viruses isolated in cell culture, can also be applied directly to the tissue; moreover, a lateral-flow rapid test (LFT) has been developed and applied to detect NNV in groupers (Nervous Necrosis Virus Rapid Test Kit, Rega Biotechnology Inc., Taipei, Taiwan). Lateral flow tests use immunoassay technology and provide very easy and rapid assays for target detection; however, antigen-detection methods have generally a limited sensitivity that makes them applicable only to cases with evident/obvious clinically affected fish and are unreliable for surveillance [48,50]. Methods based on antigen detection have been developed using both polyclonal or monoclonal antibodies [48,51,52]; due to the presence of different serotypes among betanodaviruses, specificity of these tests depends mainly on the serum used for the assay [53]. Particularly, the four genotypes have been divided into three serotypes with serotypes A matching with SJNNV, serotype B with TPNNV and RGNNV belonging to serotype C, while BFNNV position is more controversial as it has been classified as both serotype B and C [53,54].

Molecular methods applied directly to tissue samples can provide an excellent alternative to the use of cell culture for confirmatory diagnosis and for target surveillance. PCR-based methods, in fact, have short processing time, and high sensitivity and specificity, and are therefore suitable tools for the rapid detection of betanodaviruses in both clinical and subclinically infected fish. However, they need full validation through inter-laboratory proficiency tests to ascertain their performance [48].

In this regard, the OIE reference laboratory has organized several NNV Interlaboratory proficiency tests (ILPTs) to assess the capacity of laboratories to diagnose VNN using real-time RT-PCR (RT-qPCR) or endpoint RT-PCR- and to genotype detected viruses. Proficiency tests showed an overall good diagnostic capacity, but they also pointed out that there is room for improvement, particularly in the ability of laboratories to characterise different genotypes of NNV [55].

Several PCR-based techniques have been developed targeting both RNA1 and RNA2 genome segments. Despite genome-based techniques being prone to produce false negative results due to genetic variability, the selection of highly conserved genome regions can overcome this problem [56]. Conventional RT-PCR has been widely applied for confirmatory diagnosis; however, sequencing has been recommended to avoid false positive results [48].

The use of a nested-RT-PCR approach can further increase sensitivity and specificity of the test all together [29,57,58]. Comparison with conventional RT-PCR and cell culture isolation showed a 10- to 100-fold increase in sensitivity [59].

Further improvement in the use of molecular techniques to detect the virus directly in tissue samples has been provided by real time RT-PCR technology otherwise known as RT-qPCR. Particularly, these assays are less time-consuming than classical approaches, particularly in processing a large number of samples. Furthermore, real time PCR methods decrease significantly cross-contamination occurring during post-amplification procedures compared with conventional PCR. For this reason, qPCR is the molecular method proposed by OIE for target surveillance [48]. Currently, two real time PCR protocols have been fully validated by the OIE Reference Laboratory, one targeting the RNA2 [10] and the other the RNA1 [56]. The protocol of Baud et al. [56] is particularly suitable for surveillance as it can be conducted with an “on-step” approach minimizing cross contamination risks and it is highly sensitive, reaching a limit of detection of 10^2.50–2.85^TCID_50_/mL.

Some companies set up pre-mixed RT-qPCRkits for nervous necrosis virus (NNV) detection; however, they do not provide details regarding primers and probes used on their validation protocols.

Recently, the need of economic, rapid, sensitive and efficient methods to diagnose fish disease directly in the field has raised interest in techniques that require less time and experience than those required by RT-PCR-based methods. Currently, few rapid methods for the field diagnosis have been set up to detect betanodavirus, however some relevant limits with regard to their sensitivity have been raised [5]. On the other hand, biomolecular methods such as loop-mediated isothermal amplification (LAMP), cross-priming isothermal amplification (CPA) and nucleic acid sequence-based amplification (NASBA) revealed high sensitivity coupled with the advantage to conduct the whole reaction rapidly under an isothermal condition [60]. Some techniques using isothermal amplification have yet been developed and tested to detect betanodavirus in fish [61,62,63]. These methods show generally a good speed, sensitivity and cost-effectiveness; however, their efficiency in detection of different viral strains circulating in the field has not always been tested.

As a matter of fact, regardless of the technology used, the development of a ubiquitous assay detecting all viral species would be desirable, but because of the high genetic diversity of betanodaviruses it remains a big challenge. For this reason, it is essential to have a good overview of betanodavirus variant distribution to put in place the proper diagnostic tool.

Accordingly, the co-circulation of different genotypes and strains makes it particularly pressing the need for a rapid and reliable genotyping method to identify the viral genotype, which can help to uncover strain features (i.e., host range, optimal temperature) and to arrange the best control strategy [5]. Thus far, sequencing and analysis of both viral genome segments (RNA1 and RNA2) is the main method available to genotype viral strains, however this is time-consuming and in addition only few laboratories are able to apply this method to genotype the different NNV genotypes efficiently, as pointed out during the NNV Interlaboratory proficiency tests (ILPTs) [55]. Recently, a rapid identification method for NNV variants circulating in Southern Europe has been developed [64]. The method is based on an RT-multiplex PCR approach able to detect and identify RGNNV and the reassortant RGNNV/SJNNV and to distinguish them from SJNNV and SJNNV/RGNNV reassortant strains in a single RT-PCR reaction.

### Indirect Diagnosis

An indirect ELISA for the detection of anti-NNV antibodies was developed and evaluated to detect specific antibodies from naturally exposed barramundi (*Lates calcarifer*) [65,66]; median estimates of the diagnostic sensitivity and specificity calculated using RT-qPCR as a second test of the VNN ELISA were 81.8% and 86.7%, respectively. Despite the use of VNN, ELISA cannot rule out the occurrence of false positives/negatives, especially when asymptomatic fish are tested, this technique offers the advantage of non-lethal testing (particularly useful for broodstock testing), and it was fit for the purpose of identifying animals in naturally exposed populations. Moreover, with further evaluation in larger populations, this test might be used for estimating infection prevalence to facilitate risk analysis. However, further application of this technique showed the lack of association between sero-status in broodstock and the subsequent occurrence of VER disease in their progeny, indicating that ELISA tests for anti-NNV antibodies are not suitable for the purpose of preventing vertical transmission of NNV in barramundi [67].

It must be mentioned that the ELISA described by Jaramillo was an indirect assay and therefore cannot be used to test fish different from barramundi. ELISA for European sea bass has also been developed, even if its sensibility and specificity have not been reported in any paper yet. The indirect ELISA in use at the OIE reference laboratory in sea bass sera is a very sensitive technique but with average specificity. Despite some limitations, this ELISA assay has been used and is being more and more requested for research and diagnostic screening use due to its good results (Toffan, personal communication). It has been reported also that ELISA often needs to be repeated more than once to give reliable outcomes [5]. In some cases, ELISA test should be coupled with virological examination of selected fish in order to increase specificity and accuracy of the result. Different is the situation in sea bream for which no ELISA assay is currently available and the serological response against the NNV infection is almost unknown [17].

It can be concluded that the use of serology as a diagnostic/surveillance tool for antibody detection against betanodavirus is species-dependent and always needs to be integrated with other laboratory tests.

### Lethal and Non-Lethal Sampling and Sample Preservation

Lethal and non-lethal sampling methods can be used for VNN detection. Lethal samples can be easily obtained after necropsy from dead or moribund fish from clinical outbreaks (targeted samplings). For epidemiological studies (non-targeted or random sampling in large populations), euthanasia and necropsy of the sampled fish is also advisable. Lethal sampling allows wide access to all the organs and tissues and a selection of the most suitable ones. As has been previously commented, NNV mainly affects nervous tissue and eyes, so brain, spinal cord and retina are the most suitable organs or tissues to be sampled for VNN analysis. Virus can be also found in other organs such as kidney, spleen, muscle or gills but inconsistently and in a lower amount. For larvae and small fish, as necropsy and organ sampling are much more difficult to perform, whole fish (larvae) or fish heads (juveniles) are used. Lethal methods can also be used if sentinel fish are added in the different groups and used to perform routine health screenings or specific tests. The use of sentinels (usually fish kept in the rearing system for a long time) is a common strategy used in zebrafish facilities in order to detect pathologies and can also be used in Mediterranean broodstocks.

The use of non-lethal methods could be advisable in certain cases when fish euthanasia is not possible (broodstock, fish in research facilities, fish from exhibition aquaria). In these cases, non-lethal sampling can be based in blood extraction or in ovarian fluid and sperm from breeders. Blood samples can be tested for the presence of antibodies and can also be used to detect the presence of the virus using molecular techniques. However, in this case, it is important to stress that the predictive value of blood samples in the detection of the virus is lower due to problems related to the poor stability of the viral RNA in refrigerated samples, the need for specific extraction protocols and also because viruses are only present in the blood during viraemic stages. The presence of virus in non-lethal tissues including blood was inconsistent even after experimental infection [68]. Detection of antibodies against betanodavirus by ELISA (see the previous section on indirect diagnosis) and other methods were used in the past and still can be used to evaluate if the fish have been in contact with the virus and to complement other diagnostic techniques.

Sperm and mainly ovarian fluids could also be an alternative source of samples for testing by molecular methods and the presence and multiplication of the virus in the ovary and testis has been demonstrated [69] as well as vertical transmission [70]. However, there is limited information about the reliability of the use of these reproductive fluids as non-lethal samples as the extraction and detection of the virus from these samples is much more difficult due to technical reasons and the fact that the presence of virus in these fluids is not constant [70].

On many occasions, the analysis cannot be performed on site and should be performed by specialised laboratories. In this case it is very important that laboratories receive the samples in the best conditions. Live fish are always the best option but, as this is not always possible (sick fish, logistic problems), samples should be preserved and shipped in the best conditions. In any case, sample preservation should be conducted according to the requirements of each technique. Fixed samples (buffered formalin, Bouin) are recommended for histopathology meanwhile ethanol preservation, frozen samples or RNA preservative solution (the best choice to preserve RNA integrity) are recommended for molecular techniques. In any case, it is highly advisable to ask the laboratory for their preferences in the type of sample and preservation method.

## 10. Outbreak Management

Live fish are the main risk for biosecurity. As viruses necessarily require replication inside living cells of different tissues, organs and systems, infected individuals of VNN susceptible species can be considered as one of the main risks concerning prevention and biosecurity. NNV-infected fish become virus-shedders very fast as was recently demonstrated in experimental intraperitoneal and cohabitation models in juvenile seven band grouper (*Hyporthodus septemfasciatus*), the fish being able to shed large amounts of virus after 3 days and during several days post-infection [71]. The same study demonstrates in a cohabitation challenge with a proportion of 10% viral shedders on a fish group yielded 20% mortality in naïve fish, so confirming that even a very low number of fish can be sufficient to begin a new outbreak. Thus, the mere proximityof symptomatic fish can be considered as the most significant and most powerful source of large amounts of infective betanodavirus viral particles in the surrounding aquatic environment. This could be the explanation how an infection in only a very reduced number of fish in a tank or pond spreads so fast to the other fish and may explain why an outbreak in a single cage can spread to the rest of the cages of the same farm in only a few days, as is often seen in the outbreaks in the Mediterranean. Eye damage and direct release of virus to the environment from the retina could not be disregarded as a manner of facilitation for the disease. The explosive nature of betanodavirus infection can also be enhanced by several other aspects, such as an excessive number of larvae and juveniles per tank in hatcheries and nurseries, respectively, or incorrect management practices inducing non-detected underlying stress levels or higher susceptibility associated to the genetic background, amongst others. The presence of infected but asymptomatic fish in the stock before the onset of the outbreak plays a relevant role as they are shedding and spreading viruses for a certain amount of time, usually a long time, until the presence of betanodavirus virus reaches a certain threshold, becoming infective and pathogenic. This is one the more important reasons why systematic sampling schemes for betanodavirus analysis should be implemented in farms and particularly in hatcheries and nurseries. In some cases (Padrós, per. obs.), the early detection of betanodavirus in some of the batches in a facility was possible 10 days before the onset of the mortalities, providing time for the isolation of the areas where the positive stocks were detected.

To sum up, it is clear that any entrance and contact of betanodavirus within the fish stocks reared in tanks, ponds or cages in the different rearing systems (hatcheries, nurseries, ongrowing) involves a high risk of an explosive outbreak of VNN in these rearing units like a fire spark in a powder keg. Once the outbreak (still silent or evident) is generated, infected fish (diseased or not, alive or dead) become a continuous and exponential source of viruses.

Once the presence of NNV or the clinical signs of VNN have been positively diagnosed, one of the most complex problems in betanodavirus/VNN management is what should we do with the affected stock. In peracute clinical cases in hatcheries and nurseries, when progressive high mortalities suddenly appear and/or there are no doubts from clinical and laboratory results that mortalities are associated to VNN, fish in the rearing unit should be immediately sacrificed, removed from the facility and disposed in specific containers, then similar measures of disinfection and isolation should be implemented. Sometimes, farmers and staff are reluctant to sacrifice the fish believing that maybe the mortality could stop in few days and they can retrieve part of the initial stock. This is one of the most frequent mistakes, as the survivor fish, although the mortality stops, are positive and release virus for long periods of time, making them useless as they cannot be sold or transferred to other units as they are a real risk. Another important issue in these situations is the availability of very fast and reliable confirmatory laboratory results. In this case, a fast and adequate sample delivery to the laboratory and the use of fast but also validated techniques (mainly molecular) are paramount to give a clear and unambiguous diagnosis. For experienced farmers and health specialists, the concatenation of two robust positive results from the same stock sampled 5–10 days apart is enough to take the decision to sacrifice the positive stock, disinfect the area, increase the biosafety measures, and expand and increase the sampling frequency in the rest of the stock.

In hatcheries and nurseries, it is highly recommended to sacrifice immediately all the affected animals and VNN-confirmed stock. In ongrowing farms (cages or inland facilities), the strategy could be different according to the particular conditions of each farm and farming site and the expected impact of the outbreak. In farms without any previous VNN history, not vaccinated and with many risk factors present in the facility (see risk factors section), it would be advisable to sacrifice (or emergency complete harvesting) all the fish present in the affected rearing unit to try to reduce the spread of the disease to the surrounding units. In farms within VNN endemic areas, with lower risk, with vaccinated fish or with predicted low impact of the disease, a more conservative strategy could be selected.

In inland facilities, sacrificed fish (hatchery, nursery) or moribund and dead fish (ongrowing) should be removed, daily (if possible) and disposed in appropriate and labelled containers (simple and robust plastic or rubber big bin buckets with double thick big garbage plastic bags can be enough at hatchery and nursery level, larger containers such as carcass containers for terrestrial animals or large plastic insulated tubs for bigger size fish), avoiding any kind of leaking during the transport outside the facility. These bags or containers with biohazardous material should be disposed of by an authorised biological waste manager or can be temporarily stored in isolated containers or even in freezers (or in a refrigerated van or truck) with adequate biocontainment measures. In ongrowing facilities and particularly in sea cages, as the volume of dead fish can be much higher (several tons) and due to the specific conditions of these facilities, logistics are much more complicated, as dead carcasses should be manually removed by divers from the cages, as lift-up pumps are not usually available in Mediterranean farms. Collected fish should be disposed of in bigger containers (harvesting containers can be used in emergency cases), driven to the harbour and removed or stored in specific containers (hermetic industrial dumpsters) in isolated areas preventing water leaks or access to seabirds. In any case, a specific mass mortality emergency and contingency plan with the appropriate transport logistics, materials and containers and previous contacts and agreements with an authorised biological waste disposal company should be put in place (as this material can be classified as category material 1 or 2 according to regulation (EC) 1774/2002 of the European Parliament and of the Council of 3 October 2002 laying down health rules concerning animal by-products not intended for human consumption.)

After sacrifice and disposal, the fish rearing units should be physically isolated from the rest of the facility and immediately disinfected. Effluents from this rearing unit should also be disinfected accordingly. Methods and products to be used are described in the following section on biosecurity and disinfection. Obviously, isolation and disinfection mainly apply for hatchery, nursery and inland ongrowing facilities. In ongrowing in cages, for obvious reasons, isolation and disinfection measures make much less sense, as betanodaviruses can easily spread from one cage to the other cages through water currents and can remain in the environment with very few possibilities for removal in the short- or mid-term (see also the risk assessment section).

In many cases, although fast and diligent measures to remove the affected stock from the facility are taken immediately, NNV has most probably spread to other fish units (see also risk assessment and biosecurity sections). In this case, although no signs or mortalities are detected yet, it is paramount to immediately set up a surveillance plan for the rest of the units in the facility in order to assess if the virus is already present there. As, most probably the farm has different units with different fish populations, the sampling process should be conducted according to an appropriate probability sampling method, considering the characteristics (production batch, age, rearing unit) and size of each population in the facility. An adequate sampling coupled with an accurate selection of the diagnostic techniques is the best strategy to produce a reasonable assumption about if NNV has already spread to other units of the system. If results are negative, it is highly recommended to keep the rearing unit under quarantine and proceed with systematic samplings every week for at least three weeks. As was indicated before, in hatcheries and nurseries, if a specific unit tests positive, although no signs or mortality are detected, isolation, sacrifice of the stock and disinfection is indicated as the best option. If several units or batches test positive, then a complete fish stamping out and a whole disinfection of the facility should be implemented. In ongrowing, if the fish stock has achieved a certain weight and commercial value, emergency harvesting can also be considered, as VNN is not a zoonotic disease. The presence and persistence of potential virus carriers in wild fish populations around the cages should also be evaluated as these wild fish populations could act as reservoirs and vectors of the disease in the future. There is clinical and epidemiological evidence that VNN survivor second-year sea bass maintained in vicinity cages can cause a new outbreak in first-year sea bass (Zarza, personal observations). The role of asymptomatic carriers in the epidemiology of viral diseases has been recently highlighted in the COVID-19 pandemic and, therefore, exemplifies the relevance of available and accurate diagnostic techniques on the implementation of accurate diagnostic sampling strategies. These two aspects are well developed in the sections devoted to diagnostics and disease management.

After the outbreak, the production strategy in the farm should be thoughtfully reviewed according to the risks of the potential permanence of the virus in the facility. It is advisable to consider the possibility to establish a sanitary break (mainly for inland facilities but advisable also for cages), to set up a fallowing period (mainly if there is an alternative production site) and/or to start a specific vaccination program for all new batches.

## 11. Biosecurity and Disinfection

### 11.1. General Aspects

Prevention of the VNN in Mediterranean aquaculture should be visualized as all the integrative strategies designed to minimise the risk of entrance of the betanodaviruses within the different facilities, as well as the strategies designed to increase the resilience of the fish against the virus and their effects. Fish resilience against the infection and mainly immunoprophylaxis (vaccination) will be developed in the following section, but should always be seen as a complementary and synergic strategy coupled with biosecurity, mainly when biosecurity is difficult to maintain, as in ongrowing cages.

Biosecurity and disinfection for prevention of NNV infections mainly apply to broodstock, hatchery and nursery phases. In most cases, these facilities are placed in detached or modular buildings, more or less isolated from each other. Unlike salmonid farming, in Mediterranean marine farming, broodstock facilities, hatcheries and nurseries are usually placed at the same site. Only very few facilities house only broodstock and in many cases, broodstock premises are housed in the same hatchery. For betanodaviruses and other pathogen prevention, the absence of proper isolation of broodstock can be associated with a building design problem. Maybe in the past this was not a problem, as broodstock fish could be replaced easily, but nowadays this is a significant issue, as the value of the broodstock fish is increasing with the investment of genetic testing and selection in the recent years as well as the reduction in wild stock where new fish are collected. Although the risk of development of VNN with signs and mortality is relatively low compared to other rearing phases, broodstock fish in contact with betanodaviruses can become infected and stay infected for a long period of time, being a risk for the rest of the broodstock and particularly future offspring. In addition, it is confirmed that gonads are one of the organs where betanodaviruses can be detected [69,72,73]. These findings, together with the detection of the virus in recently hatched larvae, strongly support the possibility of vertical transmission of betanodavirus from broodstock [70]. In this case, the presence of betanodavirus-infected fish, even at a subclinical level, can put the whole egg and larval production at risk. For this reason, it is highly recommended to test broodstock for betanodaviruses periodically and keep, when possible, broodstock groups in isolated buildings, keeping as much isolation as possible between the different groups. If the broodstock has already been constituted, it is necessary to set up a specific testing program using non-lethal and lethal methods. Any new fish stock added to the broodstock to replace culled fish for reasons of age, male/female ratios, genetic selection or welfare issues should also be tested in a similar way, unless these fish came from a certified betanodavirus-free stock. Even to keep separate different fish species of broodstock is really important as shown by the possible role of asymptomatic carries of a species for a certain virus to another susceptible species (e.g., reassortants with sea bass and sea bream). Other practices with aspects related to risk of infection has already been described in the section on risk assessment.

For isolation, it is highly recommended to place broodstock in a separate and isolated building, not directly connected to the hatchery or to other facilities. If possible, the facility should not be placed near the seashore or near marine ponds or lagoons to reduce the risk of contact with marine water splashes, sea foam or animals living on the shoreline or backline of the coast as well as domestic animals, wild animals or pests. Entrances and exits should be designed with a double door system and should be provided with security cameras and alarm systems for non-authorised movements. Boot cleaners, usually seen in Mediterranean farms and frequently with deficient use and very poor maintenance, are not recommended for broodstock building as they do not provide sufficient biosecurity levels. If possible, showers for daily cleaning and disinfection of the staff before the entrance to the wet area are also recommended. Under no condition, external (and if possible, also internal) visits should not be authorised for biosecurity and also fish welfare reasons. As in food-processing and pharmaceutical industries, large windows placed in selected walls can allow a complete vision of the facility and can preserve isolation. A good video report about the facility is also an option, if there are not concerns on the know-how of the company.

All the equipment (nets, harvesting nets,) instruments (probes, sampling jars) and machines necessary for the operations (sampling, tagging, vaccination) and maintenance of the broodstock facility should be kept inside the building and never shared with other areas. Devices should be adequately cleaned, disinfected, dried and stored after use. It is also recommended to have dedicated staff only for the hatchery. The facility should have a specific changing room for the staff. Clean and disinfected clothing and footwear/boots should be available. The selection of a suitable biocide for disinfection according to the criteria indicate in the following section on disinfection and disinfectants is also relevant. The selection of a specific colour or design of the equipment for the broodstock facility staff is also helpful to identify and prevent risks of non-authorized movements of personnel.

Seawater supply could be obtained from different water sources: pumped from open sea, lagoons or channels, from wells or reconstituted seawater. If possible, a constant supply of good and stable water quality is desirable. However, due to the high biosecurity requirements and even if the water is pumped from wells, for safety reasons it is highly recommended to use a disinfection system that guarantees the complete disinfection before to be distributed to the broodstock. The most common disinfection systems used in Mediterranean aquaculture are combinations of filtration and UV disinfection, but seawater can also be treated using ozone, electrochemical oxidation technology or can be pre-treated with heat or by chemical strong oxidation with oxygen peroxide (see also the section on disinfection). However, chemical treatments (including ozone and electrochemical oxidation) may generate free radicals and other undesirable molecules that can be toxic mainly for larvae; here, integrating active carbon filters in the system may help to reduce this risk. In the past, flow-through systems were widely used and are still a good option but, with the development of technologically more sophisticated RAS systems, the water temperature can be maintained at a lower energy cost in these systems, and they offer an extra isolation barrier for the different stocks of breeder fish. Small- and medium-size RAS systems with reinforced disinfection systems in the inlet water, recirculating water and outlet disinfection systems are nowadays an excellent option to keep totally isolated and independent fish units with low risk of cross contamination. If freshwater is provided by the public water supply system, no disinfection is apparently necessary. Some kind of air filtration system is also desirable.

Concerning the broodstock diet, it is very important to highlight that food can also be a potential source of entrance of betanodaviruses (see VNN in the wild in the introduction section and the section on risk assessment), mainly if fish are fed fresh, refrigerated or even frozen food from the sea. In the past, chopped pieces of fish, molluscs or cephalopods were used as the main part of the diet for Mediterranean finfish species. As many fish species can be susceptible to NNV infection, or betanodavirus has been detected in these species, this practice has also been identified as a major risk for broodstock, so for biosecurity reasons it is highly inadvisable the use of pieces of fish and also bivalves or cephalopods as diet, even if they are frozen, as freezing does not destroy or inactivate NNV [74,75]. For this reason, specific broodstock feeds are the most convenient from the biosecurity point of view, as raw material processing during extrusion (high pressure, high temperature) guarantees high levels of inactivation of potential viral particles.

Collected eggs can be moved to the hatchery without many logistical problems, as the technology of eggs and larvae transport between facilities, even at long distances, has been perfectly developed. Egg disinfection with iodine or other substances such as bronopol is a common hygienic practice in several hatcheries despite it not being 100% effective against NNV; it is important to apply them for its beneficial effect preventing many other sanitary problems. There are some references on the efficacy of ozone for specific betanodavirus disinfection in the egg surface of haddock [76] but there are no specific data on disinfection efficacy and safety in eggs of seabream and seabass.

### 11.2. Biosecurity and Disinfection in Hatchery Facilities

Regarding broodstock, if possible, the hatchery should be in an isolated building and similar measures concerning the building design, equipment, assigned staff and water quality should be applied. As in broodstock, only eggs from facilities with broodstocks with specific and complete betanodavirus surveillance programmes should be accepted. As was previously indicated, detection of betanodaviruses from eggs is difficult due to technical reasons in the RNA extraction due to presence of the eggshells and the higher amount of lipids present in eggs. This is the reason why betanodavirus detection is much more feasible and reliable in larvae after the yolk sack reabsorption.

In addition to the measures previously described for broodstock, at hatchery level it is highly recommended to operate with a batch management system, as these systems allow improvement of biosecurity through the implementation of biosafety measures, pathogen firewalls and partial sanitary breaks between the different batches. Otherwise, continuous production based on daily or every 2–3 days spawnings promotes frequent regrouping and population mixing after grading, a practice that involves a high sanitary risk. A recommended system is to operate with single production batches every 10–15 days, using eggs spawned during a short period of time to keep the developmental stages/age/size of the larvae as homogeneous as possible. To do that, it is necessary to have an adequate broodstock size and a good environmental control (temperature, photoperiod) to have sufficiently controlled spawnings at the requested moment. As larvae grow, larval tanks need to be graded and moved to different and usually larger tanks; larvae from different tanks are mixed to form size-homogeneous tanks. These operations should always be conducted using pre-cleaned and disinfected tanks and immediately cleaning and disinfecting any tank and device (surface cleaners, filters and other devices) that has been used and using hygienic protocols including disinfection standards for betanodaviruses. Air supply (air blowers) in tank surface cleaners could also introduce external particles contaminated with the virus (for example sea foam) into the system, so it is highly recommended to use at least a mechanical air pre-filtration system. Batch-production systems also facilitate the introduction of partial sanitary breaks without larvae allowing more extensive cleaning and disinfection procedures in tanks, surfaces and walls.

Live food can also be considered a relevant hazard for the introduction of betanodaviruses [77]. Microalgae are sometimes used as a ‘green water’ technology to improve larval quality as the microalgae tend to stabilise and improve microbiological water quality in the tank as well as a source of valuable nutritional enrichment for rotifers and Artemia metanaupli. Microalgae can be produced as auxiliary culture in the hatchery or can also be commercially available lyophilised or as paste. In the first case, microalgal culture masters should be periodically analysed with molecular methods for the presence of betanodavirus and the water used in the system should be sterile.

Concerning rotifers, as they can be a relevant source of betanodavirus contamination [32], it is also important to keep all the master rotifer cultures under strict control to avoid contamination (water sterilisation, control of the microalgae or yeast used) and also keep a similar strict control during massive production, harvesting and enrichment. Regular control checks using PCR-based assays for viral genome detection are also recommended.

Artemia can also be a potential source of betanodaviruses and relevant attention should be paid. It is important to check the sanitary guarantee offered by the different suppliers of Artemia cysts as a relevant criterion for selection. However, additional disinfection measures are also highly recommended. If cyst decapsulation is the selected method, as high concentrations of sodium or calcium hypochlorite are usually used, this method guarantees a certain level of disinfection of any potential betanodavirus as contaminants in the chorion of the Artemia cyst. However, the absence of virus can not completely be discarded as betanodavirus could be hypothetically present in the embryo. This is why it is also recommended to include some regular checks for betanodaviruses on decapsulated eggs as part of the health/quality control of the Artemia cyst supply. As for rotifers, water supply and correct cleaning and disinfection of the materials used (filters, containers) during Artemia harvesting and eventually enrichment should be conducted.

### 11.3. Nursery and Pre-Ongrowing

In these facilities, only previously tested betanodavirus-free fish batches should be introduced. In general, due to the technical characteristics (increased biomass) and requirements of this phase, it must be assumed that the specific biosecurity measures to prevent VNN are more difficult to implement (both in technical and economic terms) than in hatcheries. Therefore, in this case it is important to reinforce the general hygienic and biosecurity protocols and implement regular samplings for betanodavirus surveillance. If fish are sold or sent to ongrowing facilities around 2 g, our recommendation is at least one sampling control and, if the pre-ongrowing period is expanded until 10–20 g, a second control in the batch is highly recommended in order to give the adequate level of guarantee that the batches are betanodavirus-free. Due to the high number of analyses performed in the different batches and especially if very sensitive techniques such as qPCR are used, implausible or unclear results are not an unusual finding. In this case, a fast reaction and re-submission of samples to the laboratory for verification or rebuttal is highly recommended.

### 11.4. Ongrowing

As ongrowing is generally performed in open systems (cages, tanks), there is a very limited possibility to apply strict biosecurity measures related to epidemiological and disease risks (see previous sections). The main strategy to be used in the reduction in the risk of potential betanodavirus outbreaks is checking that all the batches of juveniles have been previously tested negative, implementing a complete and systematic vaccination programs mainly in places with recurrent VNN outbreaks and particularly in batches with characteristics that makes them more susceptible (temperature profile, size of the stocks, etc) and establishing of an early warning system by precise and fast detection and diagnostic procedures. In areas where several farming companies coexist and share common resources (e.g., port), coordinated regional-VNN management plans should be agreed and developed for more efficient outbreak prevention and management.

The evaluation of the health quality of the juveniles and particularly the assessment and verification of betanodavirus-free status of all the batches is necessary at the level of internal quality control at farm or company level but also it is highly recommended to reinforce reciprocated confidence between fish suppliers and customers. In this mutual confidence framework, consensus is also recommended in the selection of the sampling methods and in the results of the techniques applied in these samples concerning the robustness of the results and, related to that, the decisions to be taken from these results, mainly in the face of potential positive (weak or strong) results.

As commented on in the section concerning economic impact, as the susceptibility of sea bream at large size is considered much lower than in sea bass, in case of areas with high impact of VNN outbreaks in sea bass during a long period of time, sea bream (or meagre) can be considered as an alternative to reduce the impact of the disease. However, changes associated to the emergence of new strains or reassortants could limit the efficacy of this strategy in the future.

Rearing fish in RAS conditions brings a superior level of biosecurity, particularly in the control of VNN outbreaks. Nowadays, RAS systems have started to become a real alternative for ongrowing in many species, including salmonids and particularly for Atlantic salmon. The potentially higher biosecurity standards in RAS are a very relevant factor that should be taken into consideration for the future of seabream and seabass farming, although many other factors such as initial high investment, operating costs and customer perception should be considered in the selection of the system.

## 12. General Disinfection Procedures: Systems, Chemical Products and Efficacy

To reduce the risks of entrance, permanence and spread of betanodaviruses in Mediterranean aquaculture facilities, general procedures and operations applied in HCPs (hazard control points) in different facilities (mainly broodstock facilities, hatcheries and nurseries) should be harmonised with the specific disinfection requirements for betanodaviruses.

As mentioned before, an efficient system for water disinfection of the water supply is paramount, mainly for broodstock and hatchery facilities. It is important to check that the technical specifications of the devices and systems used (filtration, UV, chemical disinfection) guarantee total inactivation of the virus potentially present in the input water. An adequate scaling of pre-filtration and filtration systems is highly recommended as this reduces the risk of entrance of viral particles adsorbed into suspended matter and, particularly, suspended in water. At the same time, pre-filtration and filtration also guarantee good water transmittance (>95%), and good transmittance is the basis for good efficiency for the UV treatments. Concerning general virus inactivation, several dose ranges have been recommended according to the different types of viruses. Some large DNA double strand viruses, such as adenoviruses, require higher doses (around 200 mW-s/cm^2^) and smaller RNA viruses such as *Picornaviridae* usually require lower doses (around 20–30 mW-s/cm^2^) [78]. Specific UV doses for betanodavirus inactivation around 100 and 300 mW-s/cm^2^ have been recommended by several authors [79,80] and by commercial websites. As the efficacy of UV systems can be limited by several factors (e.g., decreased water transmittance due to presence of very thin suspended matter, filter malfunction, problems with the water flow, technical problems with the UV devices, and mineral deposition on the surface of the UV bulbs) and in order to ensure >99.9% reduction in viable viruses, we recommend, if possible, the use of doses around 300–400 mW-s/cm^2^ system to allow higher guarantees of total inactivation of any betanodavirus particles with a wide safety margin [81,82]. General recommended disinfection procedures and protocols in the facilities and in transportation have been previously mentioned and are also described in detail [42]. Concerning specific disinfection against betanodaviruses, several doses of quaternary ammonia, iodine, acids, alkalis, formalin and chloride have been previously described by Arimoto et al. [79], Peducasse et al. [47], Frerichs et al. [80] and Maltese and Bovo [83] and are summarised in Table 1 and in Padrós et al. [42]. In the framework of the H2020 project PerformFISH, specific studies led by Prof. Sara Ciulli from the University of Bologna detail the specific disinfection activity of Virkon S against betanodaviruses in two different soiling conditions. A complete and detailed description of the experiments is summarised in Padrós et al. [42]. As a conclusion, Virkon S was effective at the manufacturer’s recommended concentration 1% *w*/*v* at both low- and high-level soiling conditions.

Thermal disinfection has also been described as an alternative treatment. Frerichs et al. [80] describe betanodavirus inactivation at 60 °C for a period of 30 min, but other authors observed higher resistance to thermal inactivation [50,82]. For certain pieces of small laboratory equipment, autoclaving is clearly a good option for disinfection, but maybe not for other equipment in hatcheries and nurseries, as they are required to be immersed in very hot water for a relatively long period of time (half an hour).

## 13. Improving Resilience: Betanodavirus Disease Resistance and Breeding Programmes

For a long time, there has anecdotal evidence regarding Mediterranean sea bass genetic lines being more resistant to NNV infection than Atlantic sea bass genetic lines. However, it was only in 2017 that Doan et al. [44] scientifically proved that the survivors after experimental infection between pure wild populations originating from Southern-East Mediterranean, Northern-East Mediterranean, Western Mediterranean and Northern Atlantic were estimated at 99%, 94%, 62%, and 44%, respectively, confirming definitively that genetics influence NNV susceptibility. Furthermore, a moderate intra-population heritability against NNV was recorded for the first time in this species, paving the way for the future NNV resistance genetic selection.

Unfortunately, in the past, the faster growing Atlantic sea bass genetics were preferred to the slower growing Mediterranean sea bass genetics and therefore massive importation and outbreeding occurred. This resulted in several mixed genetic sea bass families with an intermediate level of resistance to NNV infection. Due to the high cost of genetic analysis, only a few farms really know which genetics they possess. Indeed, the knowledge of the genetic breed by a farm can be an invaluable tool to manage the risk of NNV introduction. More recently, a specific SNP array was designed to study the resistance to VNN in three populations of European sea bass, with identification of several QTL regions involved in VNN resistance [84]. The joint effort of the MedAID and PerformFISH EU projects generated a wider tool based on the SNP technology too and validated it on over 50 seabass populations originating from 11 different Mediterranean countries (Deliverable 1.1 SNP-chip tool validation PerformFISH, [85]). This tool now can be used to characterize the genetic diversity of farmed sea bass and test their measures on inbreeding as well as for discovering genes associated with important production traits in this species. As a matter of fact, a high heritability for NNV has been already demonstrated in Atlantic cod [86,87,88] and in golden pompano [89] and therefore it will be possible in the near future even for sea bass as already partially demonstrated by several authors [90,91].

## 14. Fish Resilience: Nutritional Support

There are references in the scientific literature about anti-viral ingredients that can be potentially used in treatment of viral diseases in aquaculture [92] but, in our opinion, their commercial applicability is very limited due to the legality concerns of the use of these compounds in fish (lack of registration as a medicine or animal feed additive), availability and the potential high cost of the treatment.

On the contrary, supporting fish resilience with health diets has become an essential part of integrated preventative strategies targeting fish diseases, including viral origin, and are commonly used in fish farms. These diets have an adequate nutrient formulation and include functional ingredients, such as immunostimulants, that are capable of non-specifically increasing the natural resistance to a viral infection by enhancing or reinforcing the fish immune system; their aim is to mitigate the impact of the disease.

## 15. Immunoprophylaxis/Vaccination

The availability of vaccines to protect against VNN has always been for many years one of the main targets of researchers and the industry. Many studies at different levels have been carried out since the 1980’s and are well summarized by Costa et al. [93], Doan et al. [44] and more recently by Bandín and Souto [11]. In the past, only autologous and commercial vaccines against bacterial diseases were available for seabream and seabass, but the pressure of the continuously increasing number of outbreaks of VNN during the last two decades highlighted the need for a commercial vaccine in the market, as autologous viral vaccines (autovaccines) are only allowed in certain Mediterranean countries. Nowadays, two commercial vaccines are available and licensed for seabass in several EU countries: Alpha Ject micro®1 Noda (Pharmaq/Zoetis) and Icthiovac® VNN (Hipra), and both are based on the RGNNV genotype. Both vaccines are injectable formulations that can be applied in juvenile fish bigger than 12–15 g. The onset of the immunity is claimed to be around 450–500 degree-days (22–25 days at 20 °C) for Alpha Ject and around 920 degrees per day (46 days at 20 °C) for Ichtiovac. Protection against VNN is claimed to be for one year for Alpha Ject. Hipra vaccine displays RPS > 60% and both commercial vaccines seem to be efficient enough to reduce the mortality during outbreaks compared to non-vaccinated stocks, but they do not develop a complete protection against NNV. The availability of these two commercial vaccines can be considered a giant stride forward for the Mediterranean aquaculture and particularly for sea bass farming, as they are powerful tools used to mitigate the impact of disease, as they substantially reduce mortality and morbidity during outbreaks. These commercial vaccines, when used, protect juveniles from 15 g onwards and, after around 25–46 days after vaccination, the protection reaches its peak. As in many other vaccines, efficacy and consequently protection decrease after a certain amount of time; this means that these commercial vaccines only protect the vaccinated fish stocks at a certain level for a certain period of time. In some cases, this lasting protection can be enough to protect the whole production cycle, but not in all scenarios. In sea bass and in the Mediterranean scenario, around 400–600 days are necessary to achieve an average weight of 400 g fish and between 3 years and 3 years and 5 months to achieve 1.2 Kg (Nikos Papandroulakis, personal information). In both cases, one-year protection is clearly not sufficient to cover the whole production cycle 

With reference to sea bream, currently there are no vaccines available, but presently there is not any scientifically supported indication of the utility of an NNV vaccine in this species. As a matter of fact, biosecurity in the hatchery and ongrowing sites will be sufficient to prevent disease and mortality in this species.

Concerning broodstock and vaccines, it is important to highlight that biosecurity should be the main strategy to prevent the entrance of betanodaviruses in the system. Immunoprophylaxis (vaccination) of the broodstock fish can be a complementary strategy, but this should be considered only as a palliative measure in cases that efficient biosecurity measures could not be applied. It is important to remind that the available vaccines do not provide total (100%) protection, so broodstock fish can contraction the virus despite having been vaccinated, so vaccination cannot substitute biosecurity measures in order to prevent the entrance of the virus into the facilities.

It is worth mentioning that no data about the effect of vaccination of breeder fish on preventing viral transmission to offspring are available for sea bass nor for sea bream. The vaccination of broodstock as a protective measure has been proven on barramundi [94] and brown grouper [73], so the same could be somehow expected for Mediterranean farmed species but this needs to be confirmed before becoming an advisable procedure.

It is also important to stress that vaccination can also interfere with some diagnostic techniques as the vaccines used are not ‘marker’ vaccines. For this reason, the maintenance of labelled non-vaccinated sentinel fish could also be a useful strategy. Vaccinated fish will develop specific antibodies against betanodavirus, and this will be a problem for the interpretation of the results if diagnostics are based on antibody detection. Concerning the potential detection of viral RNA, it is important to be careful using antigen or RNA-detection-based molecular techniques, especially the most sensitive ones, during the following days after vaccination, as the remnant antigens or RNA from the vaccine could interfere during some time. In any case, if the assessment of viral RNA is conducted from brain samples, this is very unlikely.

Therefore, the key for the future improvement of the vaccination against betanodaviruses is based in the increase in the protection coverage during two specific windows: young fish and bigger fish reared in cages. Concerning young fish, it is demonstrated that the onset of almost full specific immunological capacities in seabass and seabream is around 0.5–1 g., so hatcheries and nurseries should focus on biosecurity as the main tool for betanodavirus prevention. However, as sea bass between 0.5 and 15 g are highly sensitive to betanodavirus infection in terms of outbreak risks with high mortality, there is still room for the development of vaccines and vaccination strategies that can cover this specific window of time. A first vaccination by bath at 1–2 g followed by bath or IP boosters is frequently used in some other vaccines against bacteria and could be a possibility to improve IP booster efficacy. However, it is also assumed that the onset of some levels of protection is not achieved until one month after vaccination and the level of the protection is low. IP vaccination in smaller fish could be complicated mainly due to the technical difficulties associated to fish handling, although improved vaccine formulations with lower injection doses could also help in reducing the age at first vaccination. This means that, due to these characteristics, it is expected that the immunisation coverage of future improved vaccines could be not as efficient as in larger fish.

In contrast, improvement of vaccine protection level and duration with a single dose will be very welcome in the industry. Most of the ongrowing seabass are reared in cages, frequently in environmentally exposed sites, making any kind of handling, including IP boosters, very complicated. In land-based ongrowing sites, logistics for booster vaccination are not so complicated and, maybe, if the adoption of ongrowing in RAS is also a reality in seabass and seabream farming industry, vaccination logistics could be less limited than they are now. In addition, a complete vaccination programme of seabass also requires vaccination against other pathogens (usually *Vibrio anguillarum*, *Photobacterium damselae* subs *piscicida* but also against *Aeromonas veroni*, *Aeromonas salmonicida*, *Vibrio harveyi* or *Vibrio alginolyticus* in certain endemic areas). As each IP vaccination process requires relevant logistics and the time between the minimum operational size/weight for IP vaccination before juveniles are released into production cages is very short (few weeks), there is a strong need for development and licensing of efficient multivalent vaccines including VNN in its formulation.

Alternative boosters using oral vaccines are also a possibility for development as was demonstrated in some other fish vaccines, but the relatively low level of real protection usually achieved and the technical, logistic and regulatory problems associated to the need to combine the vaccine formulation with fish feeds are a relevant hurdle for the escalation of these vaccines at commercial level.

These are the reasons why novel vaccine formulations and also novel vaccination strategies, with more robust protection levels and immunity lasting are required. New vaccines could reinforce the current scenario of vaccination for seabass and eventually also for seabream and for other species farmed in the Mediterranean if in the future the disease is expressed in bigger numbers.

## Figures and Tables

**Table 1 pathogens-11-00330-t001:** Disinfectants and doses for betanodavirus inactivation.

Substances	Quaternary Ammonia	Iodine	Acids (pH 2)	Alkalis (pH 11–14)	Formalin	Chlorine
dose/time	50 ppm / 10 min	25–50 ppm/ 30min(1) 100 ppm/ 5 min(1) 100 ppm/ 10 min(2)	15–42 days (1,2)	10 h–15 days (1,2,3)	6 h (2%) at 15 °C (1) 24 h (0,2%) at 25 °C (2) 7 days (1%) at RT (3)	25–50 ppm/ 5–30 min (1,2)
Reference	1 [79]	1 [80] 2 [83]	1 [80] 2 [83]	1 [80] 2 [83] 3 [47] 4 [79]	1 [80] 2 [82] 3 [50]	1 [80] 2 [79]

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
