# Peer review of "Integrated Management Strategies for Viral Nervous Necrosis (VNN) Disease Control in Marine Fish Farming in the Mediterranean"

_pathogens, 2022, doi:10.3390/pathogens11030330_

Round 1
Reviewer 1 Report
This is a very thorough review of management strategies for VNN control in Mediterranean aquaculture and I learned a lot reading it. The review is thoroughly referenced, a great resource for the industry and an appropriate review for this dedicated topic issue.
I felt it was logically present and thorough. The one thing I think could improve the content would be a final section that provides a summation and presents future research needs.
I did, however, spend quite a bit of time editing the English and have provided suggestions in the attached file. Due to the short time provided for editing I was unable to do more than the first 10 pages, but I think it provides examples that could be extended throughout the text. I hope this is helpful in revision, I meant for these to be suggestions only, but felt they would improve the flow of the manuscript.

Author Response
Thank you for your kind words. We are really pleased to know you enjoyed this review and we are also very grateful for the recommendations and suggestions.
Concerning the possibility of a final section, this was already considered by us, but due to the multiple topics addressed in the review and the lenght of the document, we considered that could be counterproductive and maybe overlap with the abstract. Concerning future research needs, this is precisely an exercise that will be done in PerformFISH at the end of the project (next October), so regretably and due to the deadlines of the special issue it is not possible to do it now. English edition and suggestions in the document has been added in the new version. All of them certainly improve the flow and coherence of the document. We are also very grateful for that.
Reviewer 2 Report
The manuscript “Integrated management strategies for Viral Nervous Necrosis (VNN) disease control in marine fish farming in the Mediterranean” from Pedros et al. is an impressive review on viral nervous necrosis. The authors described well the topic in many details. Nevertheless, I miss a detailed part on the virology of the Betanodaviruses, especially on the different species / types and the reassortment. Moreover, figures would be suitable for a review like this.
L113: RGNNV / SJNNV – no explanation, should be mentioned first and explained.
L779: stablished -> established
Author Response
Thank you very much for your kind words. We did our best.
Concerning your comments, we have expanded the section on virology as requested. In addition, we carefully selected and included the most relevant published references related to betanodavirus and at the same time, we did not want to repeat information or concepts that were already well described by other colleages. Amends in L113 and L779 have been included in the reviewed version.
Reviewer 3 Report
I found this manuscript really well written and I believe it will provide a complete and useful review about the knowledge and management of sanitary procedures related to NVV. I want to congratulate to the authors for such deep and well done work.
As minor point I suggest to revise once again the bibliography
since it is very wide and sometimes missed troughout the manuscript
as example I will report few point that I noticed:
Line 171-174 please add bibliography for all coinfection and particularly add confection with V. alginolyticus and association to enteropathy recently described by Savoca et al. "Savoca, S., Abbadi, M., Toffan, A., Salogni, C., Iaria, C., Capparucci, F., ... & Marino, F. (2021). Betanodavirus infection associated with larval enteropathy as a cause of mortality in cultured gilthead sea bream (Sparus aurata, Linnaeus, 1758). Aquaculture, 541, 736844."
in paragraph 7 bibliography should be added
Line 408: It should be stated that VNN is not a zoonotic disease in addition as cause not to be listed by OIE
Line 459-464: References should be added
Line 474-477 Regerences should be added
Author Response
We very much appreciate your kind words and your suggestions.
-References has been reviewed, new references has been included and amended when necessary and as suggested.
-Savoca et al. reference has been included.
-A specific reference on the fact that VNN is not a zoonotic disease has been included in the corresponding paragraph.